# When Unity Is Strength: The Strategies Used by *Chlamydomonas* to Survive Environmental Stresses

**DOI:** 10.3390/cells8111307

**Published:** 2019-10-23

**Authors:** Félix de Carpentier, Stéphane D. Lemaire, Antoine Danon

**Affiliations:** 1Institut de Biologie Physico-Chimique, UMR 8226, CNRS, Sorbonne Université, 75005 Paris, France; carpentier@ibpc.fr (F.d.C.); stephane.lemaire@ibpc.fr (S.D.L.); 2Faculty of Sciences, Doctoral School of Plant Sciences, Université Paris-Sud, Paris-Saclay, 91400 Orsay, France

**Keywords:** stress responses, acclimation, palmelloid, aggregation, programmed cell death

## Abstract

The unicellular green alga *Chlamydomonas reinhardtii* is a valuable model system to study a wide spectrum of scientific fields, including responses to environmental conditions. Most studies are performed under optimal growth conditions or under mild stress. However, when environmental conditions become harsher, the behavior of this unicellular alga is less well known. In this review we will show that despite being a unicellular organism, *Chlamydomonas* can survive very severe environmental conditions. To do so, and depending on the intensity of the stress, the strategies used by *Chlamydomonas* can range from acclimation to the formation of multicellular structures, or involve programmed cell death.

## 1. Introduction

The photosynthetic green alga *Chlamydomonas reinhardtii* is the most prominent model organism in the green algae lineage for both basic research and biotechnological applications. One of the current challenges for a better understanding of the functioning of this unicellular model organism is to understand how it adapts to hostile environmental conditions. While the different stress response strategies of multicellular photosynthetic organisms are now well known (e.g., programmed cell death (PCD), hypersensitive reaction, and autophagy) [1], much less is known for unicellular organisms like *Chlamydomonas*. Nevertheless, several reports have described the response to environmental stresses of *Chlamydomonas*, showing how this alga is able to adapt and trigger specific responses according to the intensity of the stress it faces. For instance, in *Chlamydomonas*, like in other photosynthetic organisms, environmental stresses most often result in the disturbance of photosynthesis and the production of reactive oxygen species (ROS) [2,3,4,5,6]. The first line of defense aims at containing the occurrence of this oxidative stress by decreasing ROS production and increasing ROS degradation using multiple strategies, such as non-photochemical quenching [7], alternative electron transport pathways [7,8], or the activation of antioxidant enzymes and pathways [6]. If these strategies are not sufficient to limit the impact of environmental stresses on the cell, *Chlamydomonas* will trigger alternate processes. The first strategy is acclimation which consists of building stronger defenses to protect the cell against future stress episodes. If acclimation does not provide sufficient protection to the cell, the socialization processes will be triggered, which ranges from the formation of multicellular structures to the outbreak of PCD. In all cases, interaction between *Chlamydomonas* cells will allow survival under severe stress conditions. In this review, we will describe the different strategies implemented by *Chlamydomonas* to survive environmental stresses.

## 2. Coping Strategies for Moderate Stress

### 2.1. Acclimation: Building Defenses to Protect Cells from Future Stresses

Acclimation is regarded as an environmentally-induced, short-term response leading to an improved tolerance to subsequent stresses [9]. In *Chlamydomonas*, acclimated cells are known to survive stress that would kill non-acclimated cells. Acclimation has been shown to be induced by several stresses, including ROS [10,11], UV-B [12], salt [13], and ionic stress [14]. Interestingly ROS, such as hydrogen peroxide (H_2_O_2_) or superoxide (^1^O_2_), induce specific acclimation pathways, and no cross acclimation between different ROS is observed [9,10]. Similarly, in plants, specific signaling pathways are associated with different ROS [15]. In *Chlamydomonas*, singlet oxygen acclimation is mediated by the key regulator SAK1 (singlet oxygen acclimation knocked-out 1) and the sak1 mutant is unable to acclimate to ^1^O_2_ [10]. SAK1 is a basic leucine zipper (bZIP) transcription factor that regulates the expression of important genes involved in acclimation, including *glutathione peroxidase*—*GPXH*, and *glutathione S-transferase*—*GSTS1* [10]. SOR1 (Singlet Oxygen Resistant 1) is also a bZIP transcription factor, and the corresponding mutant, which is more resistant to ^1^O_2_, has higher levels of *GPXH* and *GSTS1* transcripts [16].

It seems that at least some of the mechanisms controlling acclimation in *Chlamydomonas* are conserved in plants. For example UV-B acclimation in plants is mainly controlled by UVR8 (UV resistance locus 8) that interacts with COP1 (constitutively photomorphogenic 1) to induce a signaling pathway involving the bZIP transcription factors HY5 (elongated hypocotyl 5) and HYH (HY5 homolog) [17]. In *Chlamydomonas*, the roles of UVR8 and COP1 appear to be similar, suggesting a very early evolution of UV-B acclimation in photosynthetic organisms [12].

Acclimation is triggered in conditions of moderate stress that do not kill cells. While in the face of harsher stresses, *Chlamydomonas* will be able to enable a range of alternate strategies involving interaction between multiple cells and/or programmed cell death (Figure 1).

### 2.2. Protection by Degradation

There are two main degradative pathways to recycle cellular components in eukaryotic cells: the ubiquitin-proteasome system (UPS) and autophagy.

UPS is a degradation pathway that allows eukaryotic cells to control the abundance of numerous cellular proteins [18,19]. Ubiquitinated proteins are targeted for degradation by the proteasome, a multiproteic complex. In *Chlamydomonas*, UPS was found to be activated by stresses such as chloroplast damage [20], cold stress [21], copper [22], arsenate [23], metal nanoparticles [24], or selenite [25]. Selenite stress is associated with ROS production, which can inhibit UPS at a high concentration [25], as previously reported in mammals [26]. This could indicate that in the face of harsh stress conditions, UPS is not sufficient and more dramatic responses have to be triggered. In plants, UPS plays a prominent role in stress responses by degrading specific transcription factors [27,28]. It would be interesting to determine if such mechanisms are conserved in *Chlamydomonas*.

Autophagy is a catabolic membrane-trafficking process where cytosolic components, including proteins and membranes, are enclosed selectively or non-selectively within a double membrane vesicle, termed the autophagosome, and delivered to the vacuole or lysosome for degradation by resident hydrolases [29,30]. Autophagy usually occurs at a low, basal level under optimal growth and is activated upon stress in order to maintain cellular homeostasis. It is a pro-survival mechanism under stress conditions allowing cells to eliminate damaged cellular elements and recycle them into essential components [31]. Selective forms of autophagy exist in plants, such as chlorophagy [32] or mitophagy [33] but have not yet been reported in *Chlamydomonas*. Nevertheless, in *Chlamydomonas*, non-selective autophagy, i.e., macroautophagy (hereafter referred to as ‘autophagy’), which degrades cytosolic components, has been extensively studied during the last decade [34,35]. Autophagy is well conserved among eukaryotes and is mediated by AuTophaGy-related (ATG) genes [36]. The main role of ATG proteins is the formation and regulation of autophagosomes where damaged components will be imported. In *Chlamydomonas*, autophagy is strongly induced under different stress conditions, including nitrogen or carbon starvation [37,38], endoplasmic reticulum stress (accumulation of unfolded proteins) [39], the impairment of starch biosynthesis [40], metal exposure [41], high-light, and carotenoid deficiency [42]. The study of ATG mutants shows that autophagy is essential for cell survival, chlorophyll content, and starch accumulation during nutrient starvation [34]. ROS appears to play a major role in the control of autophagy in all organisms, including plants and algae; they have been proposed to constitute a link between the perception of stress and autophagy induction [43]. The intracellular redox state was found to control autophagosome formation and the small disulfide oxidoreductase thioredoxin plays an important role through regulation of the activity of the Atg4 protease in both *Saccharomyces cerevisiae* [44] and *Chlamydomonas* [45]. In addition, oxidative stress triggered by endoplasmic reticulum (ER) stress plays a role in the induction of autophagy [39]. Therefore UPS and autophagy are induced under moderate stress conditions, probably through ROS signaling, to eliminate toxic molecules and recycle nutrients that will allow survival until a return to optimal conditions. However, under extreme stress conditions, alternate processes are required to allow cell survival (Figure 1).

## 3. Socialization Allows for Better Resistance to Severe Stress

### 3.1. Multicellular Structures: Stronger Together

When confronted with harsh environmental stress conditions, adaptation processes are not sufficient to survive and *Chlamydomonas* cells interact to form two types of multicellular structures: palmelloids and aggregates (Figure 2). The name “palmelloid” comes from the resemblance of this structure with the alga *Palmella* [46]. Palmelloids are composed of 4 to 16 cells surrounded by a cell wall and are the result of successive divisions induced by stress without degradation of the cell wall.

In *Chlamydomonas*, stress induced palmelloid formation was initially reported by Iwasa and Murakami in the presence of organic acids [47]. Since then, a few studies have described palmelloid formation in response to a diversity of stresses (Table 1). Palmelloids can be induced by biotic and abiotic stresses in *Chlamydomonas*, and palmelloid-like structures have been reported in other green algae, such as *Scenedesmus* or *Chlorella*, suggesting that this mechanisms of stress response may be conserved in green algae [48,49].

In their natural environment, microalgae, as primary producers, are surrounded by grazers able to ingest algal cells [63]. A common defensive strategy employed by algae is to form colonies or palmelloids to exceed the upper size limit for ingestion. The rotifer predator *Brachionus calyciflorus* is indeed able to induce palmelloidation of *Chlamydomonas* within a day [50]. Palmelloid formation has also been observed under abiotic stress conditions (Table 1). During palmelloidation induced by NaCl, Khona and colleagues showed that inside the palmelloid, a stress dependent exopolysaccharides matrix is generated, probably for cell protection [55]. Inhibition of the destruction of the wall surrounding palmelloids may participate in their stabilization and could be mediated by inactivation of specific metalloproteinases (MMP) [55]. When conditions are favorable again, palmelloids dissociate rapidly; in less than an hour, and cells start to divide again (Figure 2) [55]. In case of prolonged or excessive stress, the cells remain in palmelloid form but will eventually die, through a process that may involve programmed cell death [54,55]. Proteomic analysis of the medium surrounding palmelloids indicates that the abundance of several hydroxyproline-rich glycoprotein (HPRG), such as pherophorins and MMPs, increases during palmelloid dissociation [55], suggesting that the medium could play an important regulatory role in this process. Interestingly, cells in palmelloids were found to contain more autophagic vesicles, suggesting that both autophagy and palmelloidation processes could be initiated concomitantly or that cells first induce acclimation before engaging in the path of palmelloidation [35].

In more severe conditions, the size of palmelloids may be too limited to allow survival and *Chlamydomonas* forms larger multicellular structures called aggregates or flocculates (hereafter referred to as “aggregates”). Palmelloids and aggregates are distinct structures. Aggregates are formed by a few tens, to several thousand cells, held together in an extracellular matrix, whereas palmelloids are composed of 4 to 16 cells surrounded by a cell wall (Figure 2).

Aggregation can be induced by predators, such as *Peranema trichophorum* (Euglenoidea), forming multicellular structures containing up to 100,000 *Chlamydomonas* cells [60]. Aggregated cells can secrete a mucous extracellular matrix (ECM) that may improve resistance to digestion [50,64]. Interestingly, strains of *Chlamydomonas* with smaller cells that are more easily engulfed by *Peranema* but are also slow-swimming, were found to form more aggregates compared to large fast-swimming strains [60].

The formation of aggregates may result either from agglutination of cells or from consecutive cell divisions without cell separation. When aggregation was induced in a mixed population of *Chlamydomonas* strains by the *Peranema* predator, the aggregates were found to be composed of cells from the different strains [60]. In addition, the mixing of a stained population with an unstained one also resulted in the formation of mixed aggregates [60]. This suggests that *Peranema* induces aggregation by the agglutination of cells. By contrast, aggregates of *Chlamydomonas* induced by a rotifer predator were formed only in the presence of light, suggesting that active growth may be required and that the aggregates result from consecutive divisions [50]. Therefore, multiple mechanisms of aggregate formation may exist. Unraveling these mechanisms will require more extensive cellular and molecular studies under a wide range of stresses.

Abiotic stresses, such as acidic or basic pH, can also induce aggregation (Table 1). *Chlamydomonas* culture medium is usually adjusted to pH 7. Basic pH is an efficient inducer of aggregation, especially above 10, where cations such as Ca^2+^ and Mg^2+^ form precipitates which are suggested to trigger aggregation [61]. Aggregation at low and high pH is correlated with a raise of the zeta potential [61]. The zeta potential characterizes the surface charge of a particle [65]. When aggregation is induced in *Chlamydomonas* by cationic cassia, a biopolymer, a shift of zeta potential is also observed [66]. It is, therefore, conceivable that by modifying the charge of the surface of *Chlamydomonas* cells, their ability to interact with each other would be modified and that a rise of the cell surface zeta potential might be important for aggregation.

In several studies, the mechanisms of aggregation in *Chlamydomonas* have been confronted with other species capable of forming multicellular structures. The self-flocculating yeast *Saccharomyces bayanus* was shown to induce *Chlamydomonas* aggregation, indicating that similarities in the aggregation process could exist between these two unicellular organisms [67]. In Saccharomyces, flocculation is mainly controlled by *flocculin* (*FLO*) genes [68,69] for which no homologs are present in *Chlamydomonas* genome [70]. Nevertheless, the expression of yeast *FLO5* in *Chlamydomonas* was shown to induce aggregation [71]. FLO5 is a lectin, a family of calcium dependent glycoproteins. Interestingly, concanavalin A, a plant lectin, is also able to induce aggregation in *Chlamydomonas* [67], maybe through binding to the flagella of vegetative cells [72,73]. The role of flagella seems to be important, since during aggregation induced by a predator, cell mobility was impaired although the flagella remained intact and active [74], and strains lacking flagella aggregated much less in response to pH stress [61]. Altogether these results suggest that, as in yeasts and plants, lectins could also play a role in the control of aggregation in *Chlamydomonas*, maybe in connection with flagella. Twenty genes containing a lectin-C domain have been identified in the *Chlamydomonas* genome [75] but their possible role in controlling aggregation remains to be explored.

*Volvox carteri*, a multicellular organism, belonging like *Chlamydomonas* to the order of Volvocales, uses an ECM mainly composed of HPRG to hold cells together. The heterologous expression of Volvox Algal-CAM (Cell Adhesion Molecule) is able to induce aggregation in *Chlamydomonas* [76]. Algal-CAM is a HPRG exhibiting an N-terminal extensin-like domain and two fasciclin1 (FAS1) domains. FAS1 is known to mediate cell-cell adhesion in many organisms, such as bacteria, fungi, plants, and algae [77,78]. The *Chlamydomonas* genome encodes five proteins showing identity with Algal-CAM, called FAS2, FAS3, FAS4, FAS7, and a Periostin-related protein (Cre17.g745097) [70], whose role in the aggregation process remains to be explored.

More than twenty-five examples of transitions from uni to multi-cellular organization have been reported in diverse phylogenetic groups [79]. Volvocales, with their colonial organisms like *Volvox* or *Gonium* and unicellular relatives like *Chlamydomonas*, are excellent models to study the evolutionary pathways leading from unicellularity to multicellularity [80,81]. Selection of *Chlamydomonas* sedimented cells (multicellular structures) through multiple generations allowed researchers to detect de novo multicellular forms [82]. Predation has also been used to create de novo multicellular *Chlamydomonas* lineages. Using selection in the presence of a ciliate predator (*Paramecium tetraurelia*), *Chlamydomonas*’ multicellular structures appeared after roughly 750 generations [83]. Since multicellular structures can be generated in response to specific stress conditions, it is conceivable that there is a link between stress responses, evolution, and the transition from unicellularity to multicellularity [81,84].

### 3.2. Programmed Cell Death: Sucide for the Good of the Community

Programmed cell death was identified in animals in 1972 on the basis of morphological criteria [85], and over the years, additional cellular hallmarks were found to distinguish PCD from necrosis, the accidental cell death. Mitochondria were found to play a crucial role in the control of PCD, and signals emanating from this organelle (particularly cytochrome C and apoptotic protease activating factor-1) lead to irreversible destruction of the cell [29]. This process starts with the activation of caspases, a family of proteases specifically induced during PCD. Caspases are cysteine proteases cleaving proteins after an aspartate residue, the specificity being conferred by the three amino acids preceding the aspartate [86]. The specific activities of the different caspases can be measured using artificial peptides as substrates. These peptides are often coupled to fluorophores, allowing detection of caspase activities by measuring changes in the fluorescence signal. Caspases can be inhibited using the same peptides coupled to aldehyde (CHO, reversible inhibition) or methylketone radicals (CMK, FMK: irreversible inhibition). During PCD, several caspases are activated for degradation of specific targets, resulting in DNA and nucleus fragmentation, dismantling of the cell, and the formation of apoptotic bodies [87]. The specific degradation of DNA during PCD is detectable in situ by the TUNEL (terminal deoxynucleotidyl transferase-mediated dUTP nick end labeling) reaction. It can also be visualized through gel electrophoresis by the specific appearance of a DNA ladder of 180 base pairs, the size of a nucleosome, whereas DNA degradation during necrosis is random and results in a smear [88].

Although it adopts variable forms, PCD exists in all multicellular organisms, where it is required for homeostasis, various developmental processes, and resistance to biotic or abiotic stresses [89]. If for a multicellular organism, it is clear that the disappearance of certain cells by PCD can be beneficial to the whole individual (to eliminate cells infected by a pathogen or abnormal cells for example), for a unicellular organism like *Chlamydomonas*, it seems paradoxical that PCD could exist, as it would result in the destruction of the organism itself. However, a few examples of PCD have been described in microorganisms, including, bacteria, yeasts, and microalgae [90,91,92]. In recent years, several teams have attempted to identify specific criteria for PCD in *Chlamydomonas*. In particular, it has been shown that during death induced by different stresses, caspase-like activities are induced and specific inhibitors are able to block these activities but also cell death [93,94,95]. Like land plants, *Chlamydomonas* does not appear to encode canonical caspases; however, other types of proteases have been shown to be responsible for caspase-like activities in plants [96,97]. *Chlamydomonas* contains two metacaspase genes, *MC1* and *MC2* [70]. In animals and plants, metacaspases are arginine and lysine-specific proteases which are involved in PCD induction [98,99]. This role might be conserved in *Chlamydomonas*, since *MC* genes were found to be induced during oxidative stress [100]. Therefore, the exact function of *Chlamydomonas* MC1 and MC2 will have to be clarified, for example, using knockout mutants.

The specific degradation of DNA during PCD has also been described in *Chlamydomonas* in response to different stresses, and could be detected by TUNEL or by the detection of a DNA ladder by electrophoresis [93,94,95,101,102,103]. Other markers of PCD have also been detected in *Chlamydomonas*, such as phosphatidylserine membrane translocation by annexin V [102] or fragmentation of poly(ADP-ribose) polymerase (PARP) [95]. Altogether these results suggest that a PCD-like process exists in *Chlamydomonas*, while the very reason for its existence remains to be established. Several studies have attempted to address this question by inducing PCD using heat shock [103] or UV-C [102]. In both cases the principle was to induce PCD in a culture, then to get rid of the cells to recover only the pre-conditioned “PCD” culture medium where fresh cells are inoculated, grown, and treated with the same stress that induced PCD in the first part of the experiment. The authors found that the cells placed in the PCD medium were more resistant than control cells [102,103]. Thus, the most exposed cells would be destroyed by PCD and could release into the medium, molecules capable of helping remaining cells of the population to better resist to death. This suggests that PCD should be considered as a pro-survival mechanism at the level of the whole population (Figure 1). These results also indicate that the culture medium has a fundamental role and allows cells to communicate to better adapt to their environment. The identification of molecules involved in cell to cell communications and enabling increased resistance to stress would be a major milestone for understanding this peculiar type of survival mechanism.

## 4. Cellular Reinforcement Strategies

### 4.1. The Cell Wall: The Last Defender

The cell wall plays an important role in the mechanisms allowing the protection of *Chlamydomonas* from environmental stresses. It is mainly composed of HPRGs [104], whose nature is likely to change according to environmental conditions. Consistently, cell wall-less strains are less resistant to metals such as cadmium, cobalt, copper, and nickel [105,106,107]. The cell wall has been reported to be modified by environmental conditions, including high-CO2, the presence of singlet oxygen, sulfur depravation, or an acidic pH [9,57,108,109]. High-CO2 cells produce different HPRGs and MMPs [108]; under sulfur starvation the profile of extracellular proteins is strongly modified [109,110]; and incubation at pH 3.4 induces significant thickening of the cell wall [57].

In nutrient-limiting conditions, cell wall thickening can lead to *Daphnia* resistance in *Chlamydomonas* [111]. Indeed, *Chlamydomonas* is able to resist digestion in the gut of its predator *Daphnia*, while cell wall-less strains are digested [111,112,113]. This may be crucial for survival of the population under nutrient-limiting conditions where the cell cycle is arrested, and therefore, every single life matters for the population to survive. In open-water ecosystems, a *Chlamydomonas* population could remain at equilibrium thanks to this mechanism of resistance to digestion [111,114]. It has also been proposed that the passage through the gut allows *Chlamydomonas* to take up nutrients from *Daphnia* [64]. A complete cell wall does not seem to be crucial for aggregation, as cellwall-less strains are also able to generate aggregates [61,76,115]. The role of the cell wall in the stress response has also been highlighted in cysts, where a thicker cell wall helps the cell to resist to extreme conditions.

In *Chlamydomonas*, strains used for genetic or reverse genetic experiments are usually cell-wall-deficient because of their higher transformation efficiencies [116]. Therefore special attention should be paid to studies on stress resistance using such strains.

### 4.2. Cyst: Surviving Extreme Conditions

If, like Charles Darwin, you were walking in the snows of the Andes mountains, European Alps, Antarctica, or Alaska, you might encounter the strange phenomenon of red snow [117,118]. This pink–red coloration of the snow is due to extremophile microalgae living in these hostile environments. The red snow appears in the spring/summer melting snows, at the onset of high intensities of sun radiation [119,120]. Most of snow algae are members of the Volvocales order [117]. This is a striking example of the ability of some photosynthetic microorganisms to survive under very harsh conditions, such as very low temperature, drought, starvation, and high radiation [121,122]. For instance, *Chlamydomonas nivalis* turns red in response to environmental stress, where it becomes a cyst (also named “aplanospore” or “hypnospore”) [120,121]. In *Haematococcus pluvialis*, similar encystement has been shown to be triggered by stress conditions [123], including nutrient deprivation [124], high light [125], salinity [126,127], or high or low temperatures [128]. The red pigmentation is mainly due to the accumulation of astaxanthin [129] which is produced from β-carotenes through the xanthophyll cycle using the beta-carotene ketolase (BKT) [130,131]. The very large amount of cytoplasmic astaxanthin esters acts like a filter that absorbs the excess of light that could lead to photoinhibition and photooxidative damages [121,122,132,133]. A homologue of the *BKT* gene exists in *Chlamydomonas*, although its expression level and the amount of astaxanthin are very low [134,135]; therefore, the link between BKT, astaxanthin, and stress responses, remains to be investigated. A first step may have been taken by Perozeni and colleagues, who managed to produce astaxanthin in *Chlamydomonas reinhardtii* by overexpressing a re-designed version of BKT [134].

### 4.3. Zygospore: When Sex Comes to the Rescue

In environmental stress conditions, *Chlamydomonas* haploid cells can differentiate into gametes which can mate to form diploid zygotes (Figure 2). In *Chlamydomonas*, gametogenesis have been shown to be triggered by nitrogen starvation and light variations [136], but in related organisms such as Volvox, ROS can also induce sexual behavior [137], suggesting that a greater variety of stresses could induce this differentiation in *Chlamydomonas*. The zygote-specific wall is better adapted to stress, as it is thicker and reinforced by isodityrosine cross-linking of proteins [138], and transglutamination linking lysine to glutamine [139]. Additionally, its cell wall is composed of different HPRGs [140,141], some of which are only present in the zygote, such as ZSP1 and ZSP2 (zygotic serine-proline rich) [142,143]. Recently, a giant type I polyketide synthase (PKS1) was shown to be involved in zygospore maturation [143]. The *PKS1* gene is strongly overexpressed within a few days after the mating. The pks1 mutant does not survive desiccation but is still able to germinate, suggesting that PKS1 is rather involved in stress resistance of the zygospore rather than in mating. PKS1 is mandatory for the formation of knob-like structures on the cell surface and contributes to the construction of the cell wall central layer [143,144]. This cell wall organization allows the cells to survive hostile environments, such as darkness, desiccation, starvation, and freezing [143,145,146,147]. For instance, zygospores can allow survival during the winter and can germinate when temperatures increase to release haploid cells [146]. Moreover zygospores can aggregate into clumps (Figure 2) that are extremely difficult to break up [142,144,148], and could provide additional resistance to extreme environmental stresses.

## 5. Conclusions

A better knowledge of the mechanisms controlling stress responses in *Chlamydomonas* is important both for basic research and biotechnological applications. Fundamental research may help in understanding how unicellular algae are able to withstand very unfavorable environmental conditions. We have seen specific socialization mechanisms implied, for which next to nothing is known, although this process may be central in the transition from uni to multicellularity. It would be interesting to understand which genes and signals are involved in controlling the dialogue between cells that results in multicellular structure formation but also PCD. A major goal would be to identify the molecules released into the medium by dying cells to help other cells survive unfavorable conditions. These molecules could be of great interest for many fields and applications. Understanding stress responses in *Chlamydomonas* may allow engineering microalgae for improved growth under harsh industrial conditions or high light intensities, and thereby increase the productivity and economic viability of large-scale cultures of microalgae. Cell aggregation could also prove to be a very interesting process to develop alternate and cheap methods for harvesting microalgae [149,150].

## Figures and Tables

**Figure 1 cells-08-01307-f001:**
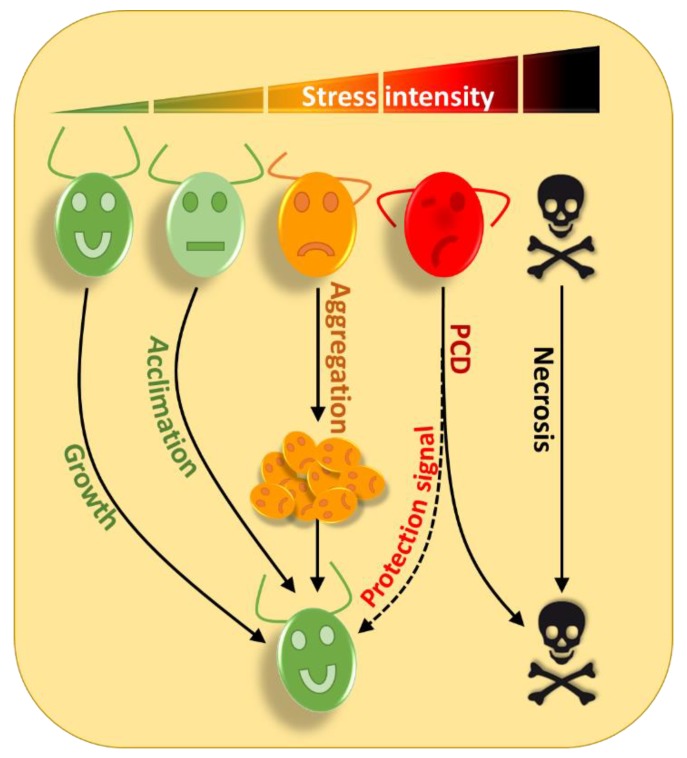
The strategies used by *Chlamydomonas* to survive increasing stress intensities. Under optimal conditions *Chlamydomonas* can grow normally (green cell); under moderate stress conditions (light green cell), several acclimation processes can be triggered to allow the cells to better adapt and resist the occurrence of any additional stress of similar or greater magnitude. Under more intense stress conditions (orange cell), cells can form multicellular structures such as palmelloids or aggregates that can dissociate when environmental conditions improve. Under very harsh stress conditions (red cell), the most exposed cells will self-destruct and release in the medium, molecules that will allow other cells to survive, to avoid the disappearance of the entire population. In the case of a stress of an intensity such that the cell cannot overcome (skull), the cell is physically destroyed by its environment through necrosis.

**Figure 2 cells-08-01307-f002:**
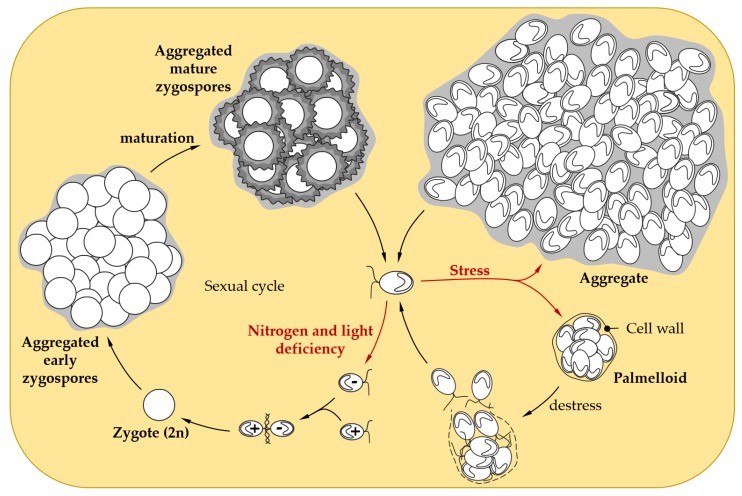
Social behavior in response to stress in *Chlamydomonas*. When facing adverse conditions, *Chlamydomonas* vegetative cells can form palmelloids, a cluster of 4 to 16 cells surrounded by a cell wall resulting from the division of a single cell. As soon as the environmental conditions improve, the outer cell wall is hatched, to allow the liberation of daughter cells [55]. When confronted with harsher stress conditions, *Chlamydomonas* cells are able to form larger multicellular structures called aggregates that might confer stress tolerance [60,61]. Under conditions of non-optimal growth (e.g., nitrogen or light limitation), *Chlamydomonas* can induce gamete differentiation to enable sexual reproduction, which leads to zygote formation after the mating of gametes. Maturation of the zygospore confers resistance to environmental stresses, such as desiccation [64]. In liquid cultures, early and mature zygospores can form aggregates, that could further enhance resistance to stress [65,66,67].

**Table 1 cells-08-01307-t001:** Multicellular structure formation in response to stress in *Chlamydomonas*. In response to different types of biotic or abiotic stresses and depending on their intensity, *Chlamydomonas* can form palmelloids or aggregates, suggesting that socialization may be a conserved mechanism that help algal cells adapt to harsh environmental conditions.

Behavior	Stress	Conditions	Reference
Palmelloids	Predator	*Brachionus calyciflorus*	[50]
	Organic acids (succinate, fumarate, aspartate, glutamate, glycolate, citrate, phthalate)	0.15–5%	[47]
	EDTA, GEDTA	1.25 mM	[51]
	Calcium deficiency	<3.5 µM	
	Phosphorous deficiency	<1 µg/L	[52]
	Cadmium	200–400 µM	[53]
	NaCl	300–700 mM	[54]
		100–150 mM	[55]
		50–150 mM	[56]
	Acidic pH	pH 4.4	[57]
	Chloroplatinic acid	50 µM	[58,59]
Aggregates	Predator	*Brachionus calyciflorus*	[50]
		*Peranema trichophorum*	[60]
	Acidic pH	pH 3.4	[57]
		pH 2.5–pH 4	[61]
	Basic pH	pH 10–pH 13	
	FeCl_3_, CaCl_2_, MgCl_2_	1–10 mM	
	Naphthenic acids	100 mg/L	[62]

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
