# Peer review of "When Unity Is Strength: The Strategies Used by *Chlamydomonas* to Survive Environmental Stresses"

_cells, 2019, doi:10.3390/cells8111307_

Round 1

Reviewer 1 Report

There is certainly room for a review on this particular aspect of Chlamydomonas biology. I enjoyed the review, especially the concept on multicellular structures.

The acclimation topic has been narrowly defined and focussing on reactive oxygen acclimation specifically. It is such a large topic that I don’t have any great suggestions other than to reiterate that acclimation responses for survival are induced in response to many different biotic and abiotic stressors. The focus on ROS makes sense in some ways since these are part of many acclimation responses. However, just the lead into this section was a little unclear. UV-B acclimation discussion seemed a bit arbitrary as well. Perhaps try to round out the acclimation section a little better and define a more central focus.

After Line 60, I found the flow improved and these more general responses were important to discuss. The autophagy discussion was an important addition to the review.

My favourite part of the review and the most unique was the discussion about socialization and stress resistance. I liked the concept and felt it was a good contribution to the field. Some discussion about losing flagella would be appropriate hear since certain stressed cells don’t swim. I though the authors explained this section quite well. Good job.

Some clarification of the cyst section should be provided. In many microalgae, you get true cyst formation as part of a “developmental” pathway. This includes cell wall thickening and changes to intracellular structures. I’m less clear on how this applies to the Chlamydomonads. The accumulation of astaxanthin could arguably be an acclimation mechanism to excess light rather than the formation of a true cyst. Perhaps add some discussion so that the meaning of cyst is clear.

The programmed cell death section isn’t that informative in the context of Chlamydomonas, but I suppose there is a limited amount of data in this field.

Specific:

Line 32—briefly describe flavodiiron proteins.

Figure 1: I like the general graphic as it sums up the discussion points in the review. The “stress intensity” scale may not be an accurate depiction as it seems to suggest a sequential order of events. This doesn’t happen in my experience, and you can clearly skip states. Perhaps the stress intensity scale is not needed.

Line 133-135: Sentence not clear. Cells do not dissociate and die?

Line 135, 259: How does the changing abundance of HPRGs relate to the composition of the media?

Figure 2: I like this one. Nice graphic.

Paragraph, Line  162-169: The zeta potential discussion isn’t clear. Try to write this paragraph so the concept is easier to follow.

Line 184-186: typo, grammatical issue.

Author Response

 The acclimation topic has been narrowly defined and focussing on reactive oxygen acclimation specifically. It is such a large topic that I don’t have any great suggestions other than to reiterate that acclimation responses for survival are induced in response to many different biotic and abiotic stressors. The focus on ROS makes sense in some ways since these are part of many acclimation responses. However, just the lead into this section was a little unclear. UV-B acclimation discussion seemed a bit arbitrary as well. Perhaps try to round out the acclimation section a little better and define a more central focus.

Line 55: We have modified this part to make clear the evocation of the UV example

 My favourite part of the review and the most unique was the discussion about socialization and stress resistance. I liked the concept and felt it was a good contribution to the field. Some discussion about losing flagella would be appropriate hear since certain stressed cells don’t swim. I though the authors explained this section quite well. Good job.

Line 180-184: We added a sentence and two references on the possible link between flagella and aggregation

 Some clarification of the cyst section should be provided. In many microalgae, you get true cyst formation as part of a “developmental” pathway. This includes cell wall thickening and changes to intracellular structures. I’m less clear on how this applies to the Chlamydomonads. The accumulation of astaxanthin could arguably be an acclimation mechanism to excess light rather than the formation of a true cyst. Perhaps add some discussion so that the meaning of cyst is clear.

Line 291:we have modified this part to focus only on the link between cyst and stress which is for now the only known example according to us

 Specific:

Line 32—briefly describe flavodiiron proteins.

Line 32: we decided to be less specific in this part and not to talk about flavodiiron proteins, as the goal is not to describe in detail the effect of this protein but to take an example.

 Figure 1: I like the general graphic as it sums up the discussion points in the review. The “stress intensity” scale may not be an accurate depiction as it seems to suggest a sequential order of events. This doesn’t happen in my experience, and you can clearly skip states. Perhaps the stress intensity scale is not needed.

We divided the stress intensity scale into several parts to make the understanding clearer

 Line 133-135: Sentence not clear. Cells do not dissociate and die?

Line 133: we hope the sentence is clearer now

 Line 135, 259: How does the changing abundance of HPRGs relate to the composition of the media?

Line 135-137: we modified the text accordingly

 Paragraph, Line  162-169: The zeta potential discussion isn’t clear. Try to write this paragraph so the concept is easier to follow.

Line 168-170: we added a sentence to try to make the concept clearer

Reviewer 2 Report

Review result of a review article by Carpentier et al.

Several factors in the stress responses have reviewed in the model alga Chlamydomonas reinhardtii. These included moderate stress response and severe stress response. The ubiquitin-proteasome system (UPS) and autophagy are involved in the protection to stress by degradation. The roles of palmelloids and aggregates in response to severe stress have also discussed. The involvement of cell wall and its components were also described.

I have found its novelty and this review covered the studies related to the stress responses.

It can be accepted for publication.

Author Response

Thank you !

Reviewer 3 Report

This review article summarizes several different responses of Chlamydomonas against environmental stress. The responses explained in the review include acclimation, palmelloid formation, aggregation, program cell death, cyst formation, and zygospore formation.

The review is well written and easy to follow the content that covers research findings from old time to most recent publications.  In principal, the article should be accepted by Cells.

The authors, however, need to add more explanation in several parts of the article before it is accepted.  In the parts, the authors simply quoted references and did not explain much. The authors seem to assume that readers are familiar with the topics or readers would obtain the references if they do not understand. This is not acceptable as a review article. 

The parts in which the authors need to add more explanation and parts the authors need to rewrite are:

Line 31: “using multiple strategies such as non-photochemical quenching [7], the recently described flavodiiron proteins [8] or the activation of antioxidant enzymes and pathways  [6].”

What are “flavodiiron proteins” and “antioxidant enzymes and pathways”.  

Line45: ROS such as hydrogen peroxide or superoxide induce specific acclimation pathways, and no cross acclimation between different ROS is observed [9,10]. Similarly, in plants, specific signaling pathways are associated with different ROS [15]. 

What are exactly “specific acclimation pathways” ?  If they are specific, please explain.  Similarly What are plant specific acclimation pathways?

Line103:  In conclusion

Why does only this section have “conclusion”?  Please remove it or add conclusion in the rest of the sections.

Line154: Aggregates induced by Peranema in a culture containing several species of Chlamydomonas are heterogenous so does aggregates with differently labeled strains from the same species [60].

The sentence not make sense. Need to explain the condition in more details.

Line165: The zeta potential characterizes the surface charge of a particle [65].

How is the zeta potential measured?

Line179: in yeasts and plants, lectins could play a role in the control of aggregation in Chlamydomonas.

The sentence not make sense.  May the authors wanted to say “lection of yeasts and plants could play a role”. Please rewrite to clarify it.

Line185: Algal-CAM

What does CAM stand for?

Line191: The transition from uni- to multicellular organization has occurred more than twenty five times in diverse phylogenetic groups [78].

The transition occurs more than 25 times in their culture periods? If so, how does the transition from multicellular to unicellular occur? If not what is it? Need more explanation.

Line197: Paramecium traurelia

Must be italic

Line265: In nutrient-limiting conditions, cell wall thickening can lead to grazing resistance [114].

Specify the algae and predator.   How much does the thickness of the cell wall increase?

Line269: Open-water ecosystems remain at equilibrium thanks to this mechanism of resistance to digestion [115]

What ecosystem is talking about? Algae species and predators? Please explain.

Line294: A first step may have been taken by Perozeni and colleagues which have managed to produce astaxanthin in Chlamydomonas reinhardtii by overexpressing BKT [134].

Does Chlamydomonas encodes and express a native BKT gene? Does non-transgenic Chlamydomonas contain astaxanthin? Please clarify.

Author Response

Line 31: “using multiple strategies such as non-photochemical quenching [7], the recently described flavodiiron proteins [8] or the activation of antioxidant enzymes and pathways  [6].”

 What are “flavodiiron proteins” and “antioxidant enzymes and pathways”.

Line 32: we decided to be less specific in this part and not to talk about flavodiiron proteins, as the goal is not to describe in detail the effect of this protein but to take an example.

  Line45: ROS such as hydrogen peroxide or superoxide induce specific acclimation pathways, and no cross acclimation between different ROS is observed [9,10]. Similarly, in plants, specific signaling pathways are associated with different ROS [15].

 What are exactly “specific acclimation pathways” ?  If they are specific, please explain.  Similarly What are plant specific acclimation pathways?

The specificity of acclimation with regard to the different types of ROS is described in the articles that we cite, we consider that it would be a little too long to develop this subject here. For the plants we speak about the signaling pathways specific to different ROS and not acclimation, once again this is described in the paper we cite.

 Line103:  In conclusion

Why does only this section have “conclusion”?  Please remove it or add conclusion in the rest of the sections.

Line 102: We modified this accordingly.

 Line154: Aggregates induced by Peranema in a culture containing several species of Chlamydomonas are heterogenous so does aggregates with differently labeled strains from the same species [60].

 The sentence not make sense. Need to explain the condition in more details.

Line 153-155: we hope that we explained it better this time

 Line165: The zeta potential characterizes the surface charge of a particle [65].

 How is the zeta potential measured?

Line 168-170: We explained better the link that could exist between the zeta potential and aggregation, nevertheless we do not think that it is the purpose of this review to explain how the zeta potential is measured.

  Line179: in yeasts and plants, lectins could play a role in the control of aggregation in Chlamydomonas.

 The sentence not make sense.  May the authors wanted to say “lection of yeasts and plants could play a role”. Please rewrite to clarify it.

Line 180: This is really what we mean, we think that this sentence makes sense.

 Line185: Algal-CAM

 What does CAM stand for?

Line 189: we have modified the text accordingly.

 Line191: The transition from uni- to multicellular organization has occurred more than twenty five times in diverse phylogenetic groups [78].

 The transition occurs more than 25 times in their culture periods? If so, how does the transition from multicellular to unicellular occur? If not what is it? Need more explanation.

Line 195-196: we have modified the text to make it clearer.

 Line265: In nutrient-limiting conditions, cell wall thickening can lead to grazing resistance [114].

 Specify the algae and predator.   How much does the thickness of the cell wall increase?

Line 270: we specified the algae and predator, nevertheless if it is clear that the thickness of the cell wall increase on the article figure, this has not been quantified.

 Line269: Open-water ecosystems remain at equilibrium thanks to this mechanism of resistance to digestion [115]

 What ecosystem is talking about? Algae species and predators? Please explain.

Line 274-275: we modified the text accordingly.

 Line294: A first step may have been taken by Perozeni and colleagues which have managed to produce astaxanthin in Chlamydomonas reinhardtii by overexpressing BKT [134].

 Does Chlamydomonas encodes and express a native BKT gene? Does non-transgenic Chlamydomonas contain astaxanthin? Please clarify.

Line 299-300: We clarified these points.